# Ancestral resurrection reveals evolutionary mechanisms of kinase plasticity

Conor J Howard[1†], Victor Hanson-Smith[2†], Kristopher J Kennedy[1], Chad J Miller[3], Hua Jane Lou[3], Alexander D Johnson[2], Benjamin E Turk[3*], Liam J Holt[1*]

[1]Department of Molecular and Cell Biology, University of California, Berkeley, Berkeley, United States; [2]Department of Microbiology and Immunology, University of California, San Francisco, San Francisco, United States; [3]Department of Pharmacology, Yale University School of Medicine, New Haven, United States

**Abstract** Protein kinases have evolved diverse specificities to enable cellular information processing. To gain insight into the mechanisms underlying kinase diversification, we studied the CMGC protein kinases using ancestral reconstruction. Within this group, the cyclin dependent kinases (CDKs) and mitogen activated protein kinases (MAPKs) require proline at the +1 position of their substrates, while Ime2 prefers arginine. The resurrected common ancestor of CDKs, MAPKs, and Ime2 could phosphorylate substrates with +1 proline or arginine, with preference for proline. This specificity changed to a strong preference for +1 arginine in the lineage leading to Ime2 via an intermediate with equal specificity for proline and arginine. Mutant analysis revealed that a variable residue within the kinase catalytic cleft, DFGx, modulates +1 specificity. Expansion of Ime2 kinase specificity by mutation of this residue did not cause dominant deleterious effects in vivo. Tolerance of cells to new specificities likely enabled the evolutionary divergence of kinases.

*For correspondence: ben.turk@ yale.edu (BET); Liam.Holt@nyumc. org (LJH)

†These authors contributed equally to this work

Competing interests: The authors declare that no competing interests exist.

## Introduction

Phosphorylation networks coordinate many cellular processes. Their importance is underscored by the prevalence of kinases: the human genome encodes >500 kinases (*Manning et al., 2002b*) and over 100,000 phosphorylation sites have been identified (*Hornbeck et al., 2012*). The number and diversity of kinases expanded with increasing numbers of cell types during the evolution of metazoa (*Manning et al., 2002a*). The addition of new kinase families with new specificities presumably increases the information processing capacity of the cell, thus enabling the emergence of more complex biological processes (*Beltrao et al., 2009*; *Lim and Pawson, 2010*).

To achieve precise regulation, kinases have evolved mechanisms to selectively phosphorylate specific substrates. This specificity is encoded at multiple levels. The active site of some kinases is optimized to bind to a defined peptide sequence, referred to as the primary specificity. Kinases may have additional peptide interaction surfaces that bind to docking motifs on the substrate distinct from the site of phosphate transfer. Non-substrate proteins called scaffolds can also form tertiary complexes to direct the interaction between kinase and substrate. The sub-cellular localization of a kinase can limit access to a subset of proteins. Finally, systems-level effects such as substrate competition and the opposing activities of phosphatases all affect the degree to which substrates are phosphorylated in the context of the cell (Reviewed in *Remenyi et al., 2005*; *Ubersax and Ferrell, 2007*).

Phosphoregulatory networks are well suited to rapid information processing because phosphorylation reactions act on time scales of minutes (*Olsen et al., 2006*). For this reason, kinase networks

**eLife digest** All living things have enzymes called protein kinases that transfer chemical tags called phosphates onto other proteins. Adding a phosphate to a protein can change the protein's activity—for example, by switching it on or off—and many biological processes involve large networks of kinases that phosphorylate hundreds of proteins.

Humans have approximately 500 different protein kinases, which can each phosphorylate many proteins, and so human cells are regulated by tens of thousands of different phosphorylation sites. This raises a number of questions: how have these different kinases evolved over evolutionary history? And how have they come to recognize, and phosphorylate, so many different sites?

Howard, Hanson-Smith et al. studied members of a large family of protein kinases called the CMGC group. These enzymes are found in all organisms that have a cell nucleus, including animals, plants, and fungi. All proteins, including kinases, are built up of a chain of smaller molecules called amino acids, and the ability of a kinase to phosphorylate a protein depends on the kinase recognizing a short string of amino acids known as a motif. The phosphate is added in the middle of this motif at the so-called '0' position. Howard, Hanson-Smith et al. found that all of the CMGC protein kinases tested 'preferred' an arginine or proline amino acid at the '+1' position of this motif. However, some kinases preferred motifs with an arginine amino acid at this position, and others preferred a proline instead.

Howard, Hanson-Smith et al. predicted how the ancestors of a number of CMGC protein kinases might have looked and then 'resurrected' them by producing them in yeast cells. When the preference of these ancestral enzymes was tested, the oldest ancestor was found to slightly prefer motifs that had a proline amino acid at the +1 position. Testing six more recent ancestors showed that, over a billion years of evolution, this amino acid preference became broader to include both proline and arginine—and that some modern protein kinases subsequently evolved and specialized to prefer arginine at the +1 position, thus creating a new specificity.

Kinases are sometimes likened to microchips in complex electronic networks. In this analogy, expanding the specificity of a kinase could be like creating many 'loose wires' and cause short-circuits. From their evolutionary analysis, Howard, Hanson-Smith et al. were able to identify a structural change in the enzyme that causes an expansion of kinase specificity, which allowed them to directly test this idea in cells. Expanding the specificity of a protein kinase that controls sexual cell division in yeast cells did not stop the yeast from dividing to produce spores, suggesting that these changes are more readily tolerated than was expected. Howard, Hanson-Smith et al. suggest that this unexpected robustness of cellular circuits enabled the evolution of the wide variety of protein kinases seen today.

are crucial for processes that require a high degree of temporal control, such as the cellular division programs, mitosis, and meiosis. Taking mitosis as an example, kinase networks control a wide range of cell sizes (2 µm to several mm) and cell biology (from a single to a thousand chromosomes, closed vs open mitosis). Thus the phosphorylation networks that underlie these processes must adapt to enable these changes in cell biology. There has been considerable progress in the understanding of transcription-factor network evolution in recent years, and these studies have helped understand the generation of morphological diversity (*Carroll, 2008*) and key principles of transcriptional rewiring (*Tuch et al., 2008*). Despite recent progress (*Holt et al., 2009*; *Tan et al., 2009*; *Alexander et al., 2011*; *Cross et al., 2011*; *Freschi et al., 2011*; *Pearlman et al., 2011*; *Capra et al., 2012*; *Lee et al., 2012*; *Beltrao et al., 2013*; *Coyle et al., 2013*; *Goldman et al., 2014*), there is still relatively little known about the evolution of kinase signaling networks.

Phosphoregulatory networks evolve by the gain or loss of protein–protein interactions, either by changes to substrates or by changes to kinase specificity. Within substrates, the gain or loss of kinase interaction motifs and phosphorylation sites have occurred relatively rapidly (changes occur within a few millions of years of divergence, *Holt et al., 2009*; *Beltrao et al., 2009*). These substrate mutations affect only one protein at a time; therefore detrimental pleiotropic effects are avoided. Alternatively, networks can evolve by changing kinase specificity. Kinases act as hubs of

phosphoregulatory networks and can coordinate the activities of hundreds or even thousands of substrates (*Manning et al., 2002a*; *Matsuoka et al., 2007*; *Holt et al., 2009*; *Hornbeck et al., 2012*). Changing the specificity of a kinase, therefore, can destroy many network connections, while also potentially creating a large number of new connections. It might be expected that there is a strong negative selection pressure against such drastic remodeling of kinase networks, but it is nevertheless clear that kinases have evolved diverse specificities, particularly following gene duplication (*Mok et al., 2010*). The mechanisms underlying this diversification are poorly understood, and it is unknown how kinases successfully evolve significant changes to their biochemistry and network biology.

To learn how kinase specificity evolves, we studied the evolutionary history of the CMGC (Cyclin Dependent Kinase [CDK], Mitogen Activated Protein Kinase [MAPK], Glycogen Synthase Kinase [GSK], and Casein Kinase [CK]) group of kinases. The CMGC group also contains the CDK-Like kinases (CDKL), SR-kinases, Homeodomain-Interacting Kinases (HIPKs), CDC-Like Kinases (CLKs), Dual-Specificity Tyrosine Regulated Kinases (DYRKs), and a paralogous superfamily of kinases including LF4, the mammalian RCK kinases (ICK, MOK, and MAK), and the fungal IME2 kinases. CMGC kinases coordinate a wide range of cellular functions in different species. CDKs are the major coordinators of cell division in both mitosis and meiosis (*Morgan, 2007*). MAPKs are crucial for many cellular decisions, including proliferation, differentiation, and stress responses (*Chen and Thorner, 2007*; *Morrison, 2012*). The Ime2 kinase is crucial for meiosis in *S. cerevisiae* (*Dirick et al., 1998*; *Benjamin, 2003*), while its orthologs in other *Ascomycetes* control distinct processes including mating (*Sherwood et al., 2014*), differentiation (*Hutchison and Glass, 2010*), and response to light (*Bayram et al., 2009*, for a review see *Irniger, 2011*). The Ime2 paralogs in mammals (the RCK kinases) control diverse processes including spermatogenesis and control of retinal cilia-length (MAK), as well as intestinal cell biology, control of cell proliferation, organogenesis, and cellular differentiation (MOK and ICK) (*Fu, 2012*).

Within the evolutionary history of CMGC kinases, gene duplications followed by diversification resulted in multiple paralogous kinases with distinct specificities that coordinate diverse biological functions. For example, the specificities of Cdk1 and Ime2 are mostly non-overlapping (*Holt et al., 2007*). In addition to acquiring distinct modes of regulation, it is likely that the divergence of the biological functions of this kinase family is, in part, due to evolution of their primary specificities. Therefore, understanding the mechanisms that drive specificity change and the consequences of these changes is crucial to rationalize the structures of modern phosphoregulatory networks. The shared evolutionary history of CMGC kinases, combined with their diverse specificities, make them an ideal gene family for studying the evolution of kinase specificity.

In this study, we determined the primary substrate specificity of eight extant kinases from the IME2/RCK/LF4 group of kinases and found variation in the amino acid that is preferred immediately C-terminal to the phosphoacceptor (the +1 position). To determine the mechanisms by which these specificities evolved, we used maximum likelihood phylogenetic models to reconstruct sequences for all ancestors of the CMGC kinases. We then resurrected seven ancestral kinases in the lineage starting with AncCMGI, which is the last common ancestor of the CDK, CDKL, MAPK, GSK, CLK, and IME2/RCK/LF4 kinases, up until the modern LF4, RCK, and IME2 kinases. Biochemical characterization of these resurrected kinases allowed us to trace the evolution of primary specificity in this lineage. In addition, we determined a key residue that modulates primary specificity at the +1 position. By mutating this residue in modern IME2 we showed that, at least in some circumstances, the cell can readily tolerate changes that expand kinase specificity.

## Results

### The Ime2/RCK/LF4 kinase family has variable +1 specificity

To understand how kinase specificity changes over a long evolutionary timescale, we determined the phosphorylation site specificities of eight kinases from the superfamily of kinase paralogs that includes fungal Ime2, the mammalian RCK kinases (ICK, MOK and MAK), and the LF4 kinases in algae and protists. This superfamily controls diverse biological processes, and we hypothesized that differences in primary specificity may underlie some of this functional divergence. In addition,

previous work has shown that *S. cerevisiae* Ime2 and mouse ICK differ in their +1 specificities (*Fu et al., 2006*; *Holt et al., 2007*).

We used a positional scanning peptide library (PSPL, *Hutti et al., 2004*) to characterize the full primary specificity of these kinases (*Figure 1A*). Briefly, we used a set of 182 peptide mixtures, in which a central phosphoacceptor position (an equal mixture of serine and threonine) was surrounded by random sequence. Within each mixture, one of nine positions was fixed to a single amino acid residue (see schematic, *Figure 1A*, top). Peptides were subjected in parallel to a radiolabelled kinase assay, and the extent of radiolabel incorporation indicates which residues are preferred or disallowed by the kinase at each position within the peptide sequence.

As previously reported for *S. cerevisiae* Ime2 and mouse ICK, PSPL analysis revealed that all kinases assayed share a strong preference for arginine at the −3 position and proline at the −2 position (*Figure 1—figure supplements 1* and *2*). However, we found that selectivity for residues C-terminal to the phosphoacceptor was more variable. Specifically, the preferred residue at the +1 position varied between arginine (+1R) for the *S. cerevisiae, Candida glabrata,* and *Yarrowia lipolytica* Ime2 homologs and proline (+1P) for the three mammalian RCK kinases. *Neurospora crassa* Ime2 and *Naegleria gruberi* LF4 phosphorylated peptides with +1R and +1P relatively equally (*Figure 1C*). All kinases also tolerated alanine (+1A) relatively equally.

Additional biochemical characterization using four consensus peptides that were varied at the phosphoacceptor (0) and +1 positions (acetyl-R-P-R-S/T-R/P-R-amide) revealed differences in steady-state kinetics underlying the +1 specificity switch. *Figure 1B* shows Michaelis–Menten curves for a pair of peptide substrates with identical sequence except for having either proline (red) or arginine (blue) at the +1 position. *S. cerevisiae* Ime2 phosphorylated the +1R peptide with 65-fold greater efficiency ($k_{cat}/K_M$) than the corresponding +1P peptide, and these differences were attributable to differences in both the $k_{cat}$ and the $K_M$ values. As anticipated, the *N. gruberi* homolog LF4 phosphorylated the +1R and +1P peptides with similar kinetics, while mouse MOK showed a twofold preference for the +1P peptide (*Table 1*).

In summary, primary specificity at the +1 position is relatively plastic among IME2/RCK/LF4 kinases, while the other major specificity determinants remained strongly conserved. Taken together with previous characterization of other CMGC kinases (*Songyang et al., 1996*, *1994*; *Fu et al., 2006*; *Holt et al., 2007*; *Sheridan et al., 2008*; *Mok et al., 2010*; *Alexander et al., 2011*; *Bullock et al., 2009*; *Kettenbach et al., 2012*), our results suggest two evolutionary hypotheses. One possibility is that the ancestor of modern CMGC kinases had dual specificity for arginine and proline and then lost either proline or arginine to specialize extant kinases. Alternatively, the +1 specificity for arginine could have evolved as a switch from proline to arginine.

## Ancestral reconstruction of the CMGC group of kinases

We sought to reconstruct the evolutionary events that led to the modern diversity of IME2/RCK/LF4 specificities. To achieve this goal, we curated a library of 329 amino acid sequences sampled broadly from across the CMGC group and then reconstructed their evolutionary history using maximum likelihood phylogenetic methods (*Thornton, 2004*; see 'Materials and methods'). The resulting phylogeny and reconstructed ancestral sequences were strongly supported by the evolutionary model (*Figure 2*, *Figure 2—figure supplement 1*). The full library of CMGC kinase ancestors is available at http://104.131.121.138/cmgc.10.2013/.

## The common ancestor of CDK, CDKL, MAPK, GSK, and IME2/RCK/LF4 (AncCMGI) had a modest +1 proline preference and was cyclin-independent

We synthesized DNA encoding the maximum likelihood common ancestor of **C**DK, CDKL, **M**APK, **G**SK, and **I**ME2/RCK/LF4 (referred to as AncCMGI). We expressed and purified the kinase from both *S. cerevisiae* and *E. coli*. We obtained active kinase in both cases. We employed positional scanning arrays to determine the primary specificity of AncCMGI, as described above and in *Figure 1*. Many kinases have fairly degenerate primary specificities (*Mok et al., 2010*) and rely on docking, scaffolding, and localization to discriminate their correct substrates (*Ubersax and Ferrell, 2007*). We therefore reasoned that an ancestral kinase might have lower specificity than the extant enzymes derived from this ancestor. Indeed, there have been several studies in which enzymes have become sub-

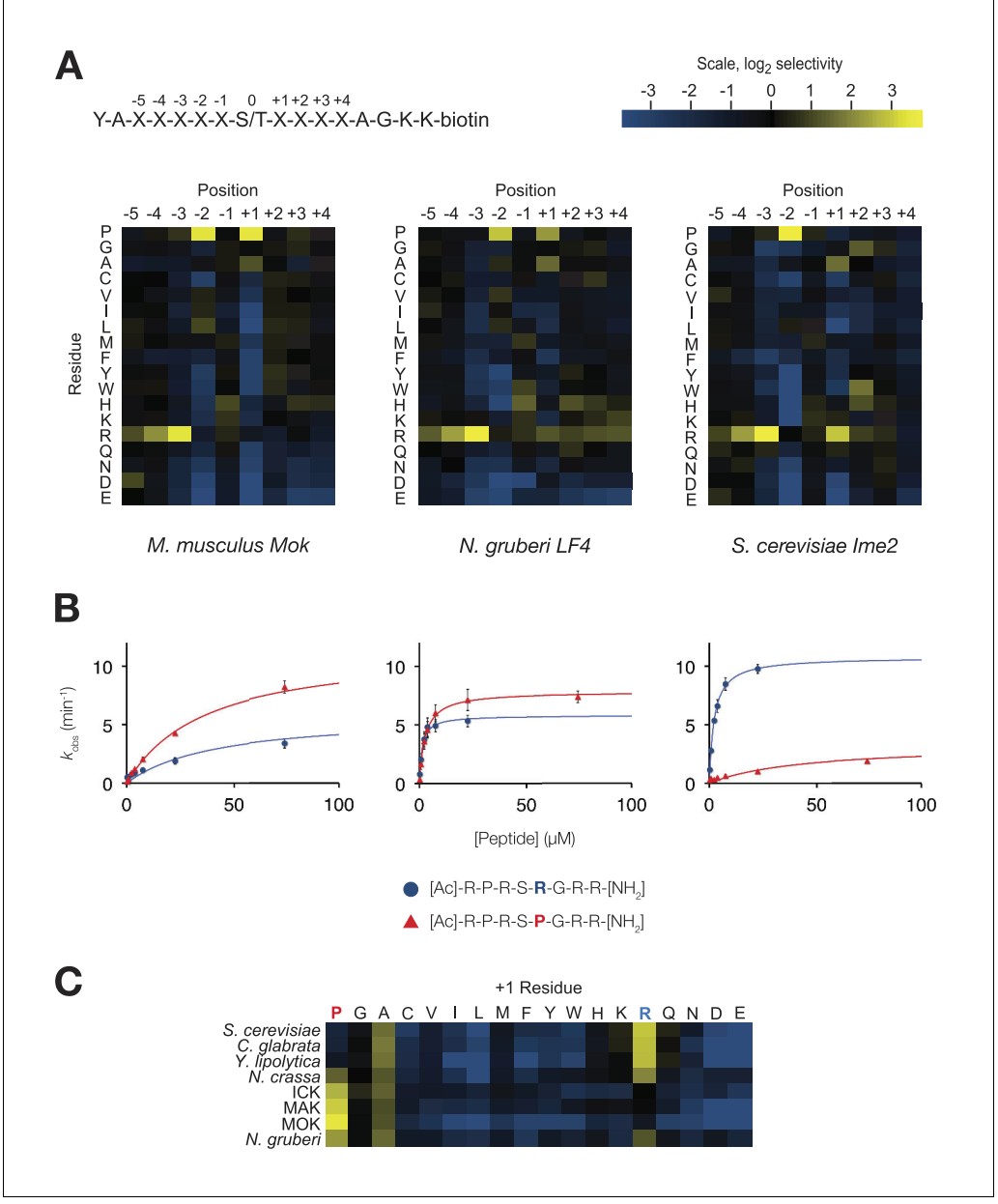

**Figure 1.** The IME2/RCK/LF4 superfamily of kinases has variable specificity at the +1 position. (**A**) Positional scanning peptide libraries were used to profile the specificity of various kinases: left, *M. musculus* MOK; middle, *N. gruberi* LF4; right, *S. cerevisiae* Ime2. Yellow indicates preference for a given amino acid while blue indicates counter selection. A schematic of the peptide library is shown above (see text for details). Data show the average of two replicates for each kinase. Raw data for these kinases and four other superfamily members are shown in *Figure 1—figure supplement 1*. Data shown here exclude peptides containing fixed Ser and Thr residues that typically produce an artificially increased signal due to the presence of an additional phosphoacceptor residue; heat maps of full peptide array results for all extant kinases are shown in *Figure 1—figure supplement 2*. (**B**) Michaelis–Menten plots for individual 8-mer IME2/RCK/LF4 consensus peptides (schematic, below) in which the +1 position is varied to be either proline (red) or arginine (blue). (**C**) Positional scanning peptide library data showing the average +1 preference of 8 kinases from the IME2/RCK/LF4 subgroup of CMGC kinases.

The following figure supplements are available for figure 1:

**Figure supplement 1.** Raw data for positional scanning peptide library arrays.

**Figure supplement 2.** Quantified positional scanning peptide library data for all extant kinases analyzed.

**Table 1.** Michaelis-Menten kinetic parameters for three members of the IME2/RCK/LF4 subgroup of kinases

| Kinase | +1 residue | $K_M$ (μM) | $k_{cat}$ (min$^{-1}$) | $K_{cat}/K_M$ (min$^{-1}$μM$^{-1}$) | |
|---|---|---|---|---|---|
| *S. cerevisiae Ime2* | R | 2.2 ± 0.3 | 10.8 ± 0.4 | 4.84 ± 0.67 | *n = 4* |
| | P | 44.5 ± 12.2 | 3.3 ± 0.3 | 0.07 ± 0.02 | *n = 4* |
| *N. gruberi LF4* | R | 1.2 ± 0.4 | 5.8 ± 0.5 | 4.76 ± 1.6 | *n = 3* |
| | P | 2.7 ± 0.6 | 7.8 ± 0.5 | 2.95 ± 0.69 | *n = 3* |
| *M. musculus Mok* | R | 41.9 ± 10.8 | 5.9 ± 0.5 | 0.14 ± 0.04 | *n = 4* |
| | P | 37 ± 2.7 | 11.7 ± 0.3 | 0.32 ± 0.03 | *n = 4* |

specialized from broad-specificity ancestors following gene duplication during evolution (*Thornton, 2003*). However, we observed that AncCMGI had a well-defined primary specificity, with a strong preference for arginine at the −3 position and proline at the −2 position (*Figure 2B*). These N-terminal determinants correspond to the conserved motif found in the IME2/RCK/LF4 kinases as well as the DYRK kinases (*Figure 2A*). Interestingly, AncCMGI could phosphorylate peptides having either a proline or an arginine residue at the +1 position, though it displayed a 5.6-fold preference for proline (*Figure 2B*). Thus AncCMGI appeared to have a modest +1 proline preference, in contrast to the more stringent proline requirement of the extant CDK and MAPK families (*Himpel et al., 2000*; *Fu et al., 2006*; *Holt et al., 2007*; *Mok et al., 2010*; *Alexander et al., 2011*). Thus, the specificity of AncCMGI contains elements of the diverged specificities of many major sub-families of the CMGC group.

Notably, the domain architecture of AncCMGI is most similar to IME2/RCK/LF4 kinases. That is, AncCMGI contains the canonical CMGC insert loop, but it lacks any C-terminal extension (found in MAPKs). Furthermore, AncCMGI does not appear to require cyclin for activity (as with CDKs): we observed no significant co-purifying proteins, and *E. coli* does not encode any cyclin orthologs (*Figure 2*, *Figure 2—figure supplement 3*). These data indicate that the cyclin dependence of CDKs and the requirement for an additional C-terminal extension to stabilize the Cα helix in MAPKs are characteristics that arose later during evolution (*E. coli* does not encode any cyclin orthologs [*Figure 2*, *Figure 2—figure supplement 3*]). In addition, AncCMGI contains a MAPK-like TXY motif in the activation loop. Phosphorylation of this motif is required for the activation of MAPKs. Because *E. coli* lack endogenous kinases capable of phosphorylating this TXY motif, AncCMGI is likely activated through autophosphorylation, similar to extant mammalian DYRK and GSK family kinases (*Cole et al., 2004*; *Lochhead et al., 2005*).

## +1 specificity evolved from modest proline preference to strong arginine preference via an expanded specificity intermediate

In order to learn the trajectory by which kinase specificity at the +1 position evolved, we reconstructed ancestral kinases within the CMGC group at multiple evolutionary time points before and after the +1 specificity change from proline to arginine was presumed to have occurred (*Figure 3A*). These kinases were assayed using consensus peptide substrates with identical sequence except for having either proline (red) or arginine (blue) at the +1 position. The log-ratio of arginine/proline preference from this assay is plotted in *Figure 3B*. Because the +1 specificity could be dependent on the surrounding sequence context present in the consensus peptide substrates, we also characterized the full primary specificities of AncLF4 and AncICK kinases by PSPL arrays (*Figure 3*, *Figure 3—figure supplement 1*). While the specificity for arginine at the −3 position and proline at the −2 position was conserved among these ancestors, we observed significant variation in their relative preference for +1R vs +1P (*Figure 3B,C*; *Figure 3*, *Figure 3—figure supplement 1*). As described above, AncCMGI appears to have preferred +1P substrates. On the phylogenetic branch leading to the common ancestor of the IME2/RCK/LF4 group (i.e., AncNgru), the +1 specificity relaxed to equally accommodate both +1R and +1P (+1PR). This hybrid specificity was conserved in the LF4 lineage and also on the branch leading to AncICK. Evolution after AncICK, however, proceeded along two divergent evolutionary paths. Namely, the specificity reverted to the ancestral +1P-preferring state along the branch leading to the mammalian RCK kinases (ICK and MAK). In contrast, the

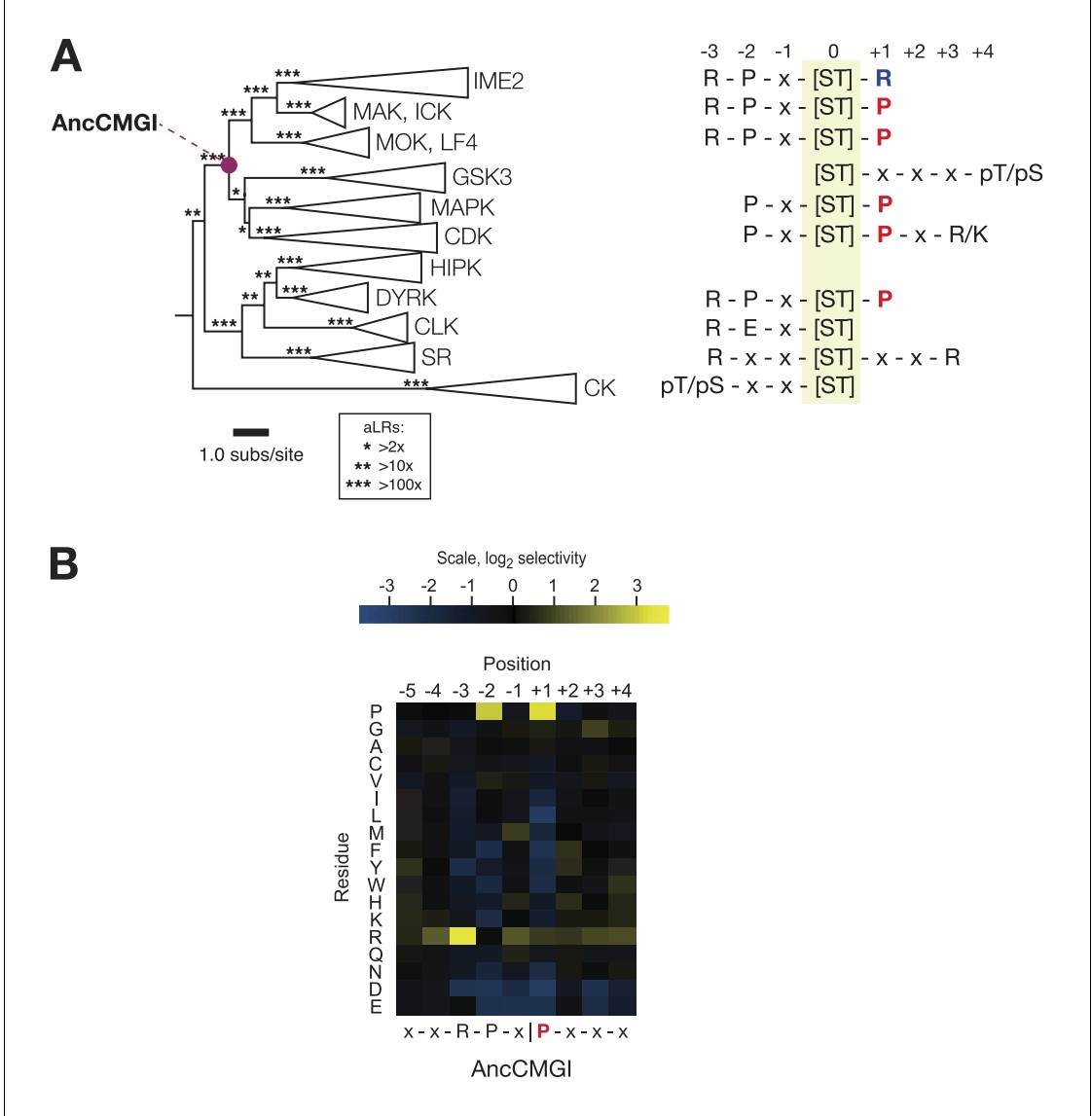

**Figure 2.** The common ancestor of CMGI kinases had a slight preference for proline at the +1 position of the substrate peptide. (**A**) Summary of current knowledge about CMGC group kinase specificity in the context of the maximum likelihood phylogeny of protein sequences. Major groups, such as IME2, MAK, ICK, etc, have been collapsed for simplicity. Branch lengths express the number of amino acid substitutions per protein sequence site. Branch support values are approximate likelihood ratios (aLRs), expressing the ratio of the likelihood of the maximum likelihood phylogeny to the next-best phylogeny lacking the indicated branch. For example, an aLR value of 10 indicates that the branch is 10 times more likely than the next-best phylogenetic hypothesis. The position of the common ancestor of CDK, MAPK, CDKL, GSK, and the IME2/LF4/RCK superfamily (AncCMGI), is indicated by a purple circle. (**B**) Position scanning peptide libraries were used to determine the primary specificity of the maximum likelihood resurrected AncCMGI kinase. Raw peptide data are shown in *Figure 2—figure supplement 1*. A complete repeat of the PSPL for a Bayesian sampled alternative reconstruction of AncCMGI (AncCMGI-B2) is shown in *Figure 2—figure supplement 2*. A structural model of AncCMGI is shown in *Figure 2—figure supplement 3* in phylogenetic context along with structures and models for extant kinases that were derived from this ancestor.

The following figure supplements are available for figure 2:

**Figure supplement 1.** Support for reconstrutctions.

**Figure supplement 2.** Raw data and selectivity values for a positional scanning peptide library array of an alternate reconstruction of AncCMGI.

**Figure supplement 3.** Structural evolution in the CMGC kinase group.

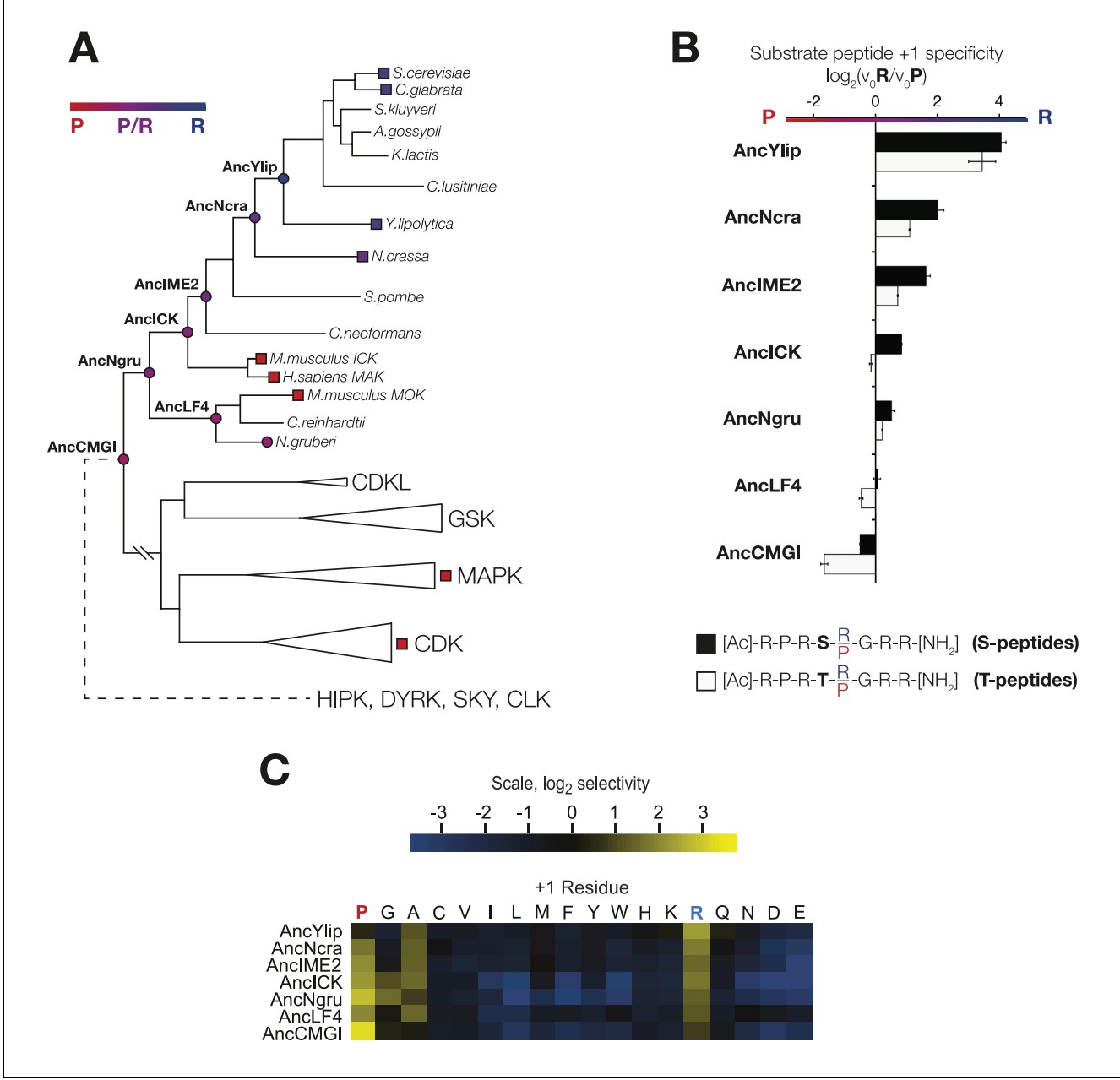

**Figure 3.** The substrate peptide +1 specificity evolved from proline in AncCMGI to arginine in *S. cerevisiae* Ime2 via an expanded specificity intermediate. (**A**) Phylogenetic tree for the IME2/LF4/RCK superfamily, also showing the positions of other major CMGC group families. The positions of ancestral nodes resurrected in this study are indicated by circles. The tree is color-coded: red indicates +1 proline preference, blue indicates +1 arginine preference, and purple indicates equal tolerance of both Arg and Pro at the +1 position. (**B**) Seven resurrected kinases were incubated with 45 μM peptide (see schematic, below). Bars show the $\log_2$ ratio of +1R and +1P initial velocities ($V_0R/V_0P$). Black and white bars indicate Ser and Thr respectively as phosphoacceptor. Error bars are standard error of three assays. (**C**) Peptide phosphorylation rates for the same resurrected kinases shown in (**B**) using the peptides from the positional scanning peptide library having the indicated residue fixed at the +1 position. Data show the average of two replicates and are normalized and $\log_2$ transformed so that the average value for a given kinase is zero. The heat map follows the same color scheme as in *Figure 1A*.

The following figure supplements are available for figure 3:

**Figure supplement 1.** Raw data and selectivity values for full positional scanning peptide arrays of AncLF4 and AncICK.

*Figure 3 continued on next page*

*Figure 3 continued*

**Figure supplement 2.** The substrate peptide +1 specificity evolved from proline in AncCMGI to arginine in *S. cerevisiae* Ime2 via an expanded specificity intermediate – robustness to uncertainty in reconstructions.

specificity shifted to +1R in the fungal lineage leading to the ancestor of the IME2 kinases (i.e., AncIME2). AncIME2 had a moderate preference for +1R, and this preference is maintained in other fungal IME2 ancestors, becoming more pronounced in the ancestor of *Yarrowia lipolytica* (AncYlip). These results are summarized in their phylogenetic context in *Figure 3A* and are robust to statistical uncertainties about the reconstructed ancestral sequences, although the degree of +1 proline selectivity was slightly lower in AncCMGI-B1 and higher in AncCMGI-B2 (see 'Materials and methods', *Figure 3*, *Figure 3—figure supplement 2*).

## The phosphoacceptor influences +1 specificity

In initial results comparing +1 specificities obtained from our positional scanning peptide library arrays to those from our ratiometric peptide assays, we noticed that ICK was an outlier among the mammalian RCKs: this kinase phosphorylated peptides with +1R and +1P equally in our ratiometric assay, while the arrays showed a clear +1P preference (albeit with detectable phosphorylation of the +1R peptide mixture, *Figure 4*, *Figure 4—figure supplement 1*). We had initially used peptides with only serine as a phosphoacceptor in our ratiometric assay, while the PSPL array peptides contained equal mixtures of serine and threonine. We therefore reasoned that the nature of the phosphoacceptor might influence the +1 specificity of kinases. To test this hypothesis, we analyzed additional peptide sets with equal mixes of serine and threonine, or with only threonine as the phosphoacceptor in our ratiometric assay. From these experiments, we found that, indeed, the phosphoacceptor affects +1 specificity: serine causes a shift towards +1R preference, while threonine causes a shift towards +1P preference (*Figure 4*, *Figure 4—figure supplement 1*). This dependence of +1 specificity on the phosphoacceptor is present in AncCMGI and is maintained in all ancestors and extant members of the IME2/RCK/LF4 family (*Figures 3B* and *4C*, *Figure 3—figure supplement 2*).

## The DFGx residue in the kinase activation loop is a determinant of +1 specificity

To understand how kinase phosphorylation site specificity changes in evolution from a structural standpoint, we sought to identify specific residues in the kinase catalytic domain that mediate +1 specificity. Kinase substrate co-crystallography and biochemical analysis of large numbers of kinases have revealed some of the rules connecting kinase sequence to specificity (*Zhu et al., 2005*; *Goldsmith et al., 2007*; *Mok et al., 2010*). The peptide-binding groove is formed from a number of structural elements within the kinase catalytic domain. One key point of kinase-substrate interaction is the activation loop, a conformationally flexible region that extends between two highly conserved amino acid motifs, DFG, and APE, connecting the N- and C-terminal kinase lobes (*Huse and Kuriyan, 2002*). Previous work revealed that the amino acid immediately C-terminal to the conserved DFG motif contributes to preference for serine vs threonine at the phosphoacceptor (*Chen et al., 2014*). This residue was previously referred to as the DFG+1 residue, but here we will refer to this amino acid as the **DFGx** residue to avoid confusion with the +1 amino acid position of the substrate peptide. Since the DFGx residue communicates with the phosphoacceptor, and the phosphoacceptor in turn influences +1 specificity, we hypothesized that the identity of the DFGx residue may be a determinant for +1P vs +1R specificity. This hypothesis is consistent with X-ray crystal structures of kinase peptide complexes, in which the +1 residue in the substrate is in close proximity to the DFGx residue (*Soundararajan et al., 2013*) (*Figure 4A*). Our ancestral reconstructions indicate that the DFGx residue was a leucine in AncCMGI and then mutated to serine multiple times in the CMGC family (*Figure 4B*). Further, the presence of leucine vs serine at DFGx in present-day kinases correlates with specificity for +1P vs +1R. Taken together, this suggests that Leu vs Ser at DFGx could affect kinase specificity at the +1 site. To test this hypothesis, we examined the effect of mutating Leu to Ser in mammalian ICK (L146S). This single mutation at DFGx made ICK approximately threefold more proline specific, such that its specificity resembled that of MOK (*Figure 4C*). Conversely,

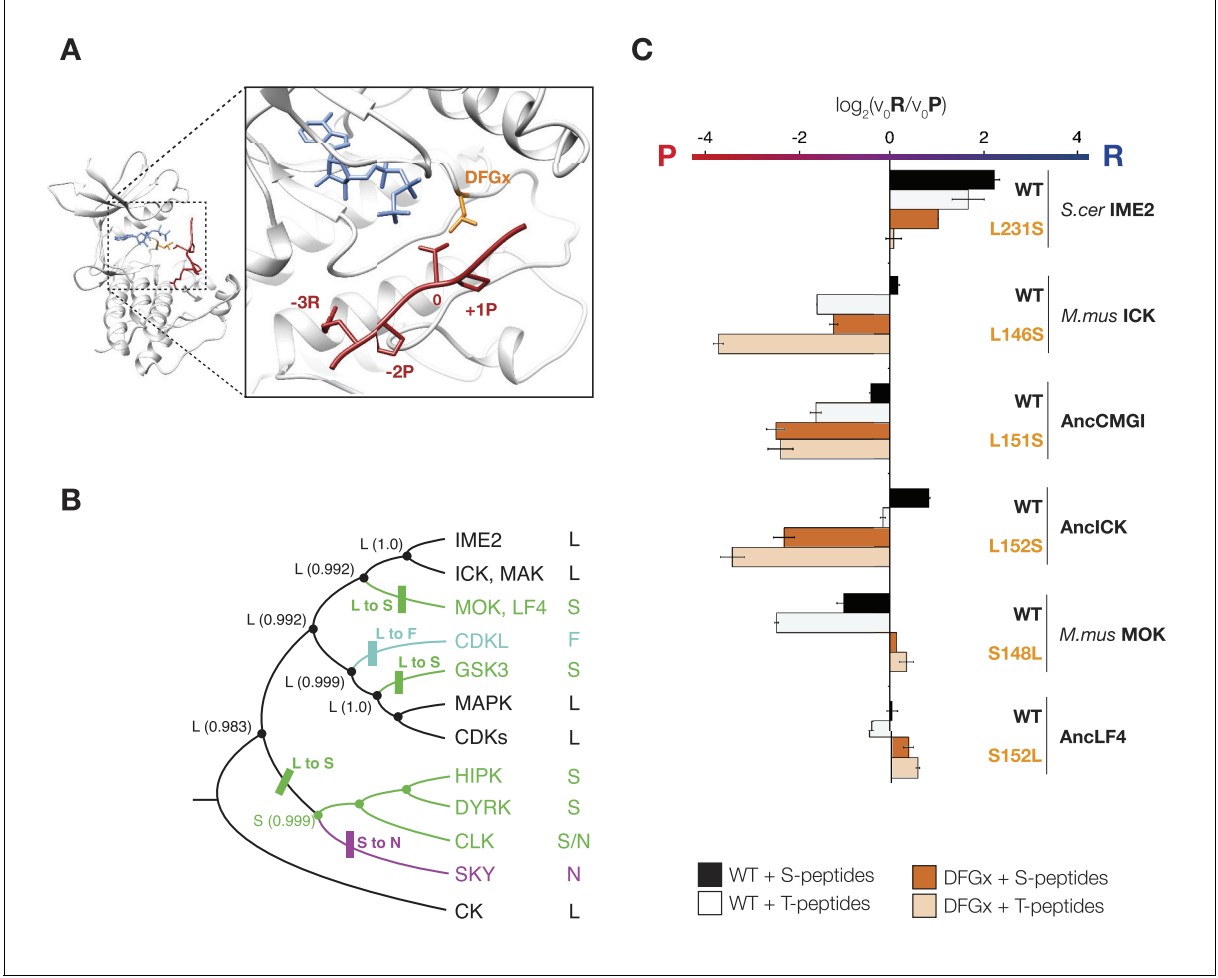

**Figure 4.** The DFGx amino acid and the phosphoacceptor influence the +1 specificity of extant and ancestral kinases. (**A**) Structural model of AncCMGI in complex with a consensus peptide substrate. The box shows the active site with the position of the DFGx amino acid highlighted in orange. ATP is blue and the substrate peptide is red. For clarity, sidechains are only shown for residues discussed in the text. (**B**) Phylogenetic tree indicating the identity of the DFGx amino acid and the transitions that occurred in the evolution of the CMGC group. Numbers indicate support for ancestral reconstructions. (**C**) Kinases were incubated with 45 µM peptide and initial velocities measured. In general, L to S mutations shift substrate preference towards +1P while S to L mutations shift preference towards +1 R. Bars show the $\log_2$ ratio of +1R and +1P initial velocities ($V_0R/V_0P$). Black and white bars indicate wild type or maximum likelihood kinases incubated with peptides that contain serine and threonine respectively as phosphoacceptor. Dark and light orange indicate DFGx mutant kinases incubated with peptides that contain serine and threonine respectively as phosphoacceptor. Error bars are standard error of three assays. *Figure 4—figure supplement 1* shows data for ICK compared to PSPL results. *Figure 4—figure supplement 2* shows full Michaelis–Menten curves for selected kinases and DFGx mutants.

The following figure supplements are available for figure 4:

**Figure supplement 1.** The phosphoacceptor affects the +1 specificity of ICK.

**Figure supplement 2.** Michaelis–Menten curves for selected kinases.

**Figure supplement 3.** Variable phosphoacceptor preference for extant IME2/RCK/LF4 kinases.

**Figure supplement 4.** Phosphoacceptor preference shows a general shift from threonine towards threonine during evolution in the IME2/RCK/LF4 lineage.

mutating the DFGx residue in the opposite direction in MOK (S148L) had the reverse effect of converting MOK from +1P preference to a non-selective kinase (+1PR). We also mutated the DFGx residue in *S. cerevisae* Ime2 and in the ancestors AncLF4(S152L), AncICK(L152S), and AncCMGI(L151S). In all cases, we observed that mutation from leucine to serine at DFGx shifted kinase function towards +1P specificity, while mutation from serine to leucine at DFGx shifted the kinases towards +1R specificity (*Figure 4C*). We note that though DFGx mutation was sufficient to substantially shift the +1 residue preference, other residues must also be important for +1 specificity, since proline to arginine selectivity shifts occur in the evolution of CMGC kinases without a DFGx mutation.

In addition to affecting kinase selectivity at the +1 position, as anticipated mutation of the DFGx residue also impacted phosphoacceptor preference (see *Figure 4*, *Figure 4—figure supplement 2*). However, the identity of the DFGx residue appeared to only modestly affect the phosphoacceptor specificity in comparison to specificity at the +1 position, and did not follow a clear pattern. These results are in keeping with previous observations that CMGC kinases are generally less specific than other kinase groups for the phosphoacceptor residue (*Chen et al., 2014*; *Figure 4*, *Figure 4—figure supplement 3*). We also examined phosphoacceptor preference in our resurrected CMGC ancestors. A general trend is observed in which AncCMGI has a slight preference for serine and this shifts towards a slight threonine preference in the lineage leading to Ime2 (*Figure 4*, *Figure 4—figure supplement 4*).

## The *S. cerevisiae* meiotic phosphoregulatory network tolerates expansion of kinase specificity

In *S. cerevisiae*, Ime2 is expressed exclusively during meiosis and is required for all stages of this process, including meiotic initiation, S-phase, the meiotic divisions, and gametogenesis (*Yoshida et al., 1990*; *Dirick et al., 1998*; *Benjamin, 2003*; *Holt et al., 2007*; *McDonald et al., 2009*). This meiotic exclusive expression allows us to engineer allelic replacements of *IME2* without any impact on vegetative cells, and then assess the ability of strains to complete various aspects of the meiotic program.

The Ime2 DFGx mutant shifts from a strong +1R preference to an expanded specificity that tolerates both proline and arginine at the +1 position. This mutant has a comparable turnover to the wild-type kinase (*Figure 4*, *Figure 4—figure supplement 2*) and therefore can be used to meaningfully test the effect of changing primary specificity on phosphoregulatory networks in vivo. To this end, we replaced the endogenous *IME2* gene with the *ime2-DFGx* allele and assayed the ability of cells to undergo meiosis.

As reported previously (*Benjamin, 2003*), a kinase-dead version of Ime2 (*ime2-K97R*) failed to support meiosis (not shown). However, cells with both copies of *IME2* replaced with a DFGx mutant (*ime2-L231S*) completed meiosis, but with a reduction of sporulation efficiency thus indicating that the *ime2-DFGx* allele has significant activity in vivo (*Figure 5A*). The homozygous *ime2-DFGx* cells that did correctly form tetrads had reduced spore viability (*Figure 5B*, *Figure 5—figure supplement 1*), and initiated S-phase (*Figure 5C*) and the meiotic divisions (*Figure 5D*) with a 2–3 hr delay. These defects may be caused by a weakening of network connections due to a reduced preference for arginine at the +1 position in the DFGx mutant.

The defects in the homozygous *ime2-DFGx (L231S)* strain could be due to either a weakening of existing network connections, a gain of new network connections that interfere with meiotic processes or a combination of both. We reasoned that the loss of important phosphorylation events would be recessive and could be compensated for by the presence of a wild-type *IME2* allele, while the gain of new phosphorylation sites would be dominant. To test for dominant effects in the DFGx mutant, we generated heterozygous *IME2/ime2-DFGx* strains. *IME2/ime2-DFGx* strains had no noticeable defects in meiosis or sporulation. DNA replication and divisions occurred on schedule, spore formation was normal, and spore viability was high (*Figure 5*). Therefore, we conclude that the defects seen in the homozygous DFGx mutant strains are most likely due to weakening of network connections, rather than the creation of new deleterious phosphorylation sites. This experiment is analogous to gene duplication followed by paralog divergence in evolution and indicates that the meiotic phosphoregulatory network can tolerate divergence of specificity in a second copy of the *IME2* gene.

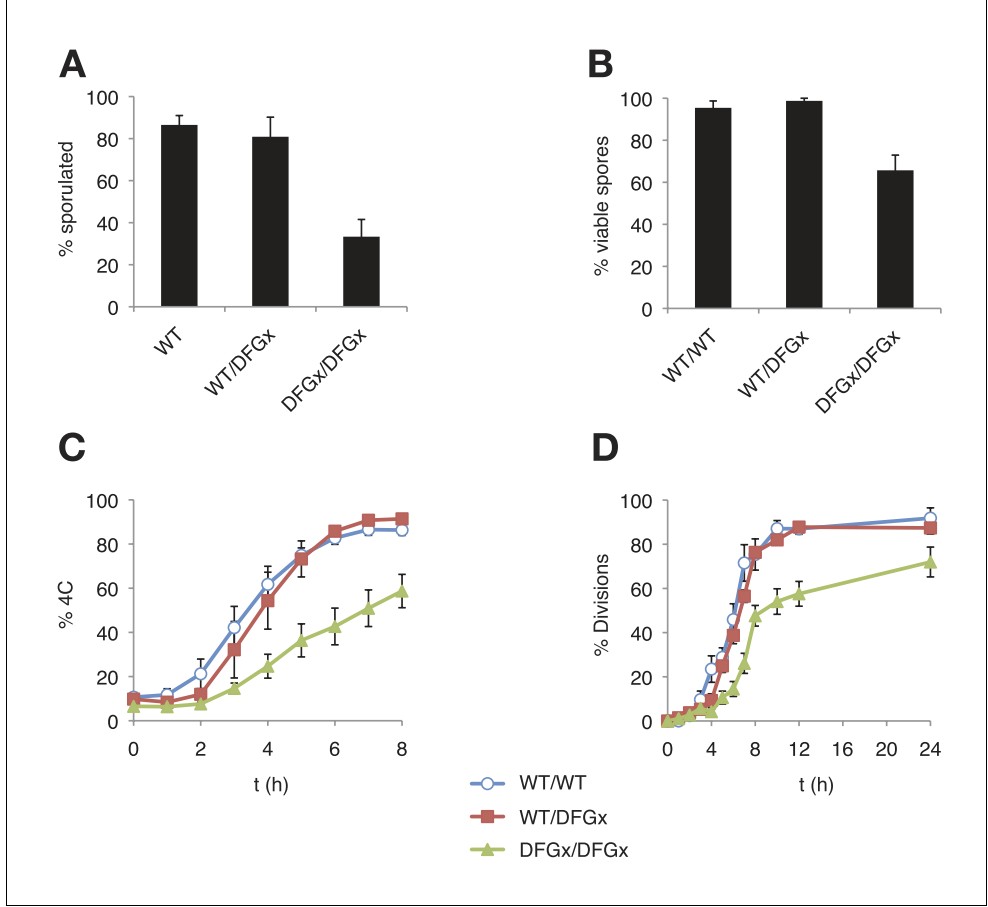

**Figure 5.** The *S. cerevisiae* meiotic phosphoregulatory network tolerates an expanded specificity DFGx mutant. (**A**) Sporulation efficiency with various *IME2* alleles: wild-type *IME2* (WT), an *ime2-(L231S)* heterozygote (WT/DFGx), or an *ime2-(L231S)* homozygote (DFGx/DFGx). (**B**) Fraction of tetrad spores that, when dissected, gave rise to colonies (representative images shown in ***Figure 5—figure supplement 1***). (**C**) Synchronous meiosis was induced and DNA content analyzed by SYTOX-Green staining and flow cytometry (representative raw data shown in ***Figure 5—figure supplement 2***). (**D**) Synchronous meiosis was induced and DNA segregation events were scored by fluorescence microscopy. Error bars represent standard error of three or more biological replicates.

The following figure supplements are available for figure 5:

**Figure supplement 1.** Representative tetrad dissections.

**Figure supplement 2.** Representative cytometry data from meiosis experiments.

## Discussion

We resurrected multiple ancestral CMGC kinases and characterized their specificities in order to reveal the evolutionary trajectories by which specificity at the +1 site in present-day kinases diversified. The ancestor of all CMGC kinases had slight specificity for proline at its +1 site (+1P), and then subsequently evolved broader specificity for both proline and arginine (+1PR) along the lineage leading to mammalian RCK and fungal IME2 kinases. After the (+1PR) hybrid ancestor, evolution followed various trajectories. In mammalian RCK kinases, the specificity reverted to the ancient modest proline specificity, while in *N. gruberi* LF4 and *N. crassa* Ime2 the equal preference for proline and arginine was maintained. In *S. cerevisiae, Y. liplolytica,* and *C. glabrata*, the hybrid intermediate kinase evolved towards arginine specificity (+1R). Our molecular manipulation experiments showed that substitutions at a single residue in the active site of the kinase sequence, at the DFGx site, had a significant effect in determining +1P vs +1R specificity. Finally, by studying kinase function in vivo,

we showed that cells tolerate the hybrid +1PR specificity. Taken together with the measured specificities of reconstructed CMGC kinases, these results suggest that the evolutionary trajectory from +1P to +1R passed through a historic ancestor with hybrid +1PR specificity. However, due to the modest degree of +1 specificity in AncCMGI (5.6-fold by PSPL and 2-fold in the context of an otherwise perfect consensus peptide) together with slight fluctuations in degree of specificity in alternate reconstructions, an alternative possibility is that AncCMGI was a broad specificity kinase that subfunctionalized to modern proline or arginine specific kinases.

Our results are consistent with previous phylogenetic studies suggesting that IME2 kinases are closer to the ancestral state and that CDKs are more derived (*Krylov et al., 2003*). In fact, we observed that AncCMGI, the maximum likelihood common ancestor of **C**DK, CDKL, **M**APK, **G**SK, and **I**ME2/RCK/LF4, has essentially the same specificity as the mouse MOK kinase and does not require a cyclin or any C-terminal extension for its activation. Our phylogenetic reconstructions indicate, therefore, that cyclin dependence is a derived characteristic. Although our study focused on the evolutionary trajectory leading from AncCMGI to the IME2/RCK/LF4 superfamily, we in fact reconstructed ancestral sequences in all lineages of the CMGC tree, including the ancestors leading to mammalian CDKs and MAPKs. This library of ancestral kinases is a rich resource that can be used to elucidate the evolutionary paths by which MAPKs and CDKs acquired their unique forms of allosteric regulation.

Selectivity for proline at the +1 position is a unique characteristic of the CMGC group, as kinases from other groups appear to strongly disfavor proline (*Zhu et al., 2005*). This proline selectivity has been attributed to an arginine residue found at the C-terminus of the activation loop (at the xAPE position) that is unique to the CMCG group. In structures of CMGC kinases, this arginine residue interacts with the backbone carbonyl group of another residue in the activation loop, termed the 'toggle residue', to orient it away from the bound substrate. In contrast, in other kinase groups the toggle residue carbonyl is oriented toward the substrate, forming a hydrogen bond with the amide proton of the +1 residue. Thus, the presence of arginine at the xAPE position is thought to explain the +1 preference of CMGC kinases for Pro, a secondary amino acid lacking an amide proton. However, because all CMGC kinases have arginine at this position, and not all CMGC kinases prefer proline at the +1 position, other residues must be responsible for our observed differences in substrate specificity among members of this group. We have identified the DFGx residue as one such determinant. Our ancestral reconstructions predict several additional amino acids as determinants of +1 specificity, and future work will illuminate their effects.

Computational analyses have suggested that kinases display little inter-positional dependence in their primary peptide specificity and that each substrate peptide amino acid interacts with the kinase active site more or less independently (*Joughin et al., 2012*). Our observation that +1 specificity is dependent on the identity of the phosphoacceptor represents an exception to this model. This inter-positional dependence is ancestral and has been maintained in the entire IME2/RCK/LF4 family of kinases. The common role for the DFGx residue in influencing specificity for both the +1 and phosphoacceptor residues provides a structural basis for inter-dependence between the two positions. It is likely that similar inter-positional dependencies exist in other kinase families in cases where the two residues share an overlapping binding site. For example, crystal structures of tyrosine kinase–peptide complexes have revealed a single binding pocket accommodating both the +1 and +3 residues (*Bose et al., 2006*). Such information should be considered when designing future phosphorylation site prediction algorithms.

Previous studies have shown that specificity-shifting mutations tend to lead to large losses in enzyme activity. As such, the evolution of new specificity often proceeds by first acquiring permissive amino acid substitutions that stabilize the protein conformation and then next acquiring substitutions that shift specificity. For example, in the case of the glucocorticoid receptors, it was inferred that neutral mutations that stabilized a new conformation must have been acquired before the specificity-shifting mutation could arise in the receptor's active site (*Ortlund et al., 2007*). In another example, it was shown that permissive mutations were required in influenza neuraminidase prior to acquisition of drug resistance mutations that subtly altered binding specificities (*Bloom et al., 2010*). In directed evolution experiments, additional compensating mutations were required to restore wild-type levels of activity to proteases mutated in their specificity pockets (*Varadarajan et al., 2008*). This requirement for multiple epistatic mutations is likely to slow the evolution of specificity in these systems, while also significantly reducing the chance of convergent

specificity evolution. The evolution of CMGC kinase specificity at the +1 site, however, is an outlier to this paradigm. CMGC kinases seem to be relatively tolerant to modulation of +1 specificity by mutation of the DFGx residue: this mutation did not lead to a significant loss of activity in any of the six kinases we tested. This is an unusual case where a single amino acid mutation can drive divergence of specificity without the need for additional stabilizing mutations. Perhaps this tolerance explains the repeated convergent evolution of the DFGx residue.

There has been considerable evolutionary diversification to the primary specificities of the CMGC kinases such that paralogs diverged by more than two billion years have almost no overlap in their preferences. In order to make progress studying a tractable period of evolutionary change, we focused our analysis on the sub-family rooted at the common ancestor of Cdk1 and Ime2 (i.e., AncCMGI). Following AncCMGI, a gene duplication occurred, and the specificities of the two descendant paralogs almost completely diverged. Cdk1 recognizes a [S/T*]-P-x-[K/R] motif, while its paralog Ime2 recognizes R-P-x-[S/T*]-R (*Figure 2A*; *Holt et al., 2007*). We studied the mutational trajectory of the paralog leading to Ime2, and a similar analysis awaits for the paralog leading to Cdk1. The mechanism by which both paralogs were retained is unclear, although previous work has revealed how other gene paralogs were retained according to various evolutionary behaviors, including sub-functionalization (*Force et al., 1999*; *Finnigan et al., 2012*), neo-functionalization (*Bridgham, 2006*; *Conant and Wolfe, 2008*), and avoidance of paralog interference (*Baker et al., 2013*). Future work studying the Cdk1 paralog will reveal to what extent these models fit the evolution of CMGC kinases.

The evolution of phosphoregulatory networks is analogous to the evolution of gene transcription regulatory networks (*Moses and Landry, 2010*). In both cases, changing the specificity of the regulator gene—either a kinase or a transcription factor—can potentially lead to the loss of essential network connections and also potentially create a large number of new connections (*Figure 6*). The gain or loss of connections might be a barrier to evolution if the changes were to interfere with existing biology. Nevertheless, it is clear that both transcription regulators and kinases have undergone extensive diversification in their specificities and network connections. Although much has been written about the evolution of transcription regulators, there has been comparatively little work on the evolutionary patterns and processes of kinases and phosophoregulatory networks. In light of the fact that the DFGx mutation shifts specificity from +1R to an expanded +1RP without a loss of activity, this allele provides a model for us to investigate whether dominant effects might constrain phosphoregulatory network evolution. The specificity modulation in the DFGx mutant could in theory generate hundreds of new phosphorylation events and modulate the rates of many more. However, our initial genetic analysis of the DFGx mutant suggests that expansion of Ime2 specificity is not highly toxic to cells. As long as a wild-type copy of IME2 is present, the second copy can expand its activity without any noticeable reduction of meiotic efficiency.

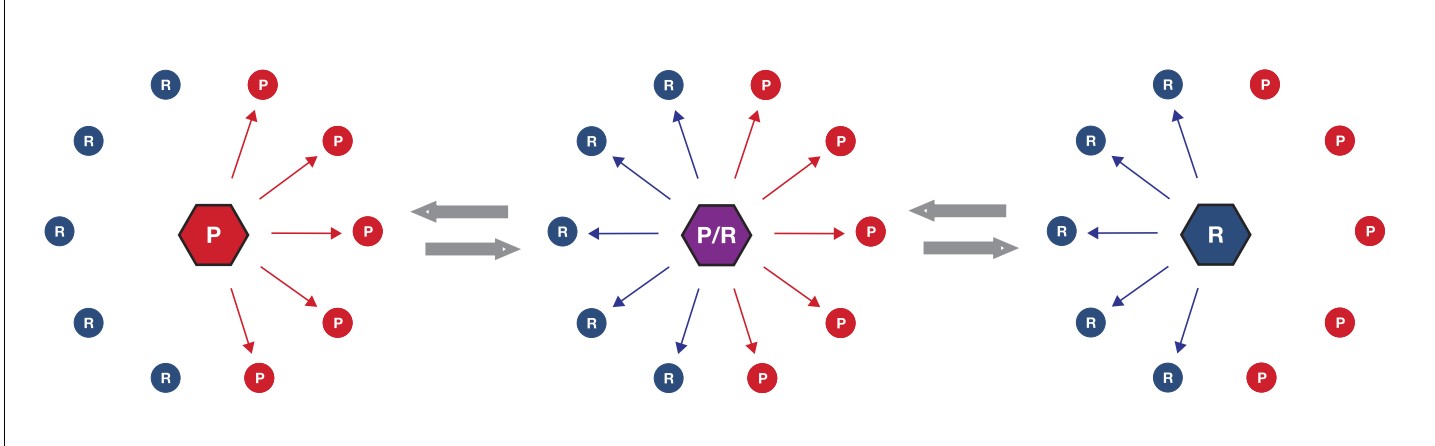

**Figure 6.** Simplified schematic of one possible path of kinase network evolution. Hexagons represent kinases, circles represent substrates, lines connecting kinase to substrate indicate potential phosphoregulation.

There are several possible explanations for the tolerance of the meiotic network to the DFGx heterozygous state. First, it could be that the modulation of primary specificity does not lead to a large number of phosphor transfer events to novel peptides. This would be the case if secondary specificity determinants such as docking interactions, scaffolding, or localization were dominant over primary specificity. For example, if the kinase was constrained to a subset of proteins by strong docking interactions, the potential to phosphorylate other proteins might be limited. Second, it could be that new phosphorylation events are occurring but that they have mild or insignificant biological consequence (*Landry et al., 2009*). It remains an open question what proportion of the tens of thousands of phosphates detectable in the cell serve to modulate substrate function: perhaps it is a minority. Thirdly, it could be that phosphatases in the cell dominate over the novel phosphorylation events from the DFGx mutant kinase. In this case, phosphates would be transferred to new substrate peptides, but phosphatases would quickly remove them, such that steady-state substrate phosphorylation levels are low. Genetic experiments with phosphatase mutants could be informative here, and these experiments are ongoing.

In conclusion, we combined ancestral reconstruction and functional biochemistry to elucidate part of the mechanism by which the ancestor of the CDK and IME2 kinase families switched from a slight +1P to +1R specificity. This specificity change is likely to have evolved via an expanded specificity intermediate, suggesting that phosphoregulatory networks are robust enough to tolerate the introduction of new specificities. We have obtained initial evidence that the extant network that controls meiosis in *S. cerevisiae* can tolerate the introduction of the expanded specificity *ime2-DFGx* allele without a large loss of fitness. Evolution of new kinases is likely to begin with the chance generation of new specificities that give some advantage. However, to ensure an evolutionary future for new kinases, novel substrate connections need to arise and the balance with competing phosphatases must be tuned to generate advantageous networks that support the ever-elaborating diversity of life.

## Note added in proof

During the process of submission the authors have learned that the DFGx specificity mutation described in this study is also observed in multiple cancer kinases including Aurora A, PKCgamma, Haspin, DDR1, ITK, TRKA, IRAK3 and BRAF (Creixell et al. unpublished), suggesting that amino acid changes that occur during evolution are resampled in cancer. It will be interesting to see if these mutations cause specificity changes in the context of oncogenesis.

## Materials and methods

### Plasmids

Kinases were either cloned by PCR from respective organisms or from gifts: *N. crassa* Ime2 was a gift from Louise Glass, ICK and MOK cDNA were gifts from Tom Sturgill and Zheng Fu (*Fu et al., 2006*), MAK cDNA was a gift from Alex Bullock, *N. gruberi* genomic DNA was a gift from Lillian Fritz-Laylin (*Fritz-Laylin et al., 2010*). Ancestral kinases were synthesized either from gBlock gene fragments (IDT) or by Genscript. All plasmids were assembled by Gibson isothermal assembly (*Gibson et al., 2009*), cloned in *E. coli* XL1-blue strains, and prepared by miniprep (Qiagen, Redwood City, CA). The list of plasmids used in this study is presented in *Supplementary file 1*.

### Clarification of mammalian RCK family gene names

The RCK family of kinase is identified by various synonyms in the literature. Therefore, to avoid confusion, the mammalian RCK kinases used in this study are:

1. **ICK** *Mus musculus* (MGI:1934157, Mouse synonym: 2210420N10Rik; Human orthologue ICK, HGNC:21219, Human synonyms: **I**ntestinal **C**ell **K**inase, KIAA0936, LCK2, MGC46090, MRK);
2. **MOK** *Mus musculus* (MGI:1336881, Mouse synonyms: **M**APK/MAK/MRK **O**verlapping **K**inase, MOK, Rage, Stk30; Human orthologue MOK, HGNC:9833, Human synonyms: Renal Tumor Antigen, RAGE1, STK30);
3. **MAK** *Homo sapiens* (HGNC:6816, **M**ale germ cell-**A**ssociated **K**inase, Human synonyms: dJ417M14.2, RP62).

## Yeast strains

Yeast strains were generated by standard transformation and crossing protocols. Protein purification was performed from W303 strains. We initially performed meiotic experiments in SK1 strains derived from the Herskowitz collection (*Benjamin, 2003*), but later switched to SK1 strains from Angelika Amon (a gift from Elçin Ünal). Both SK1 strains gave similar results but the Amon background was more consistent. All yeast strains were generated by standard LiAc transformation (*Amberg et al., 2005*). SK1 and W303 strains were heat shocked at 42°C for 15 and 40 min respectively. Point mutations of *IME2* in the SK1 background were generated by 2-step loop-in, loop-out gene replacement technique using selection and counter-selection of the *URA3* marker at the *IME2* genomic locus. The list of yeast strains used in this study is presented in *Supplementary file 2*.

## Synchronous sporulation timecourses

Liquid sporulation was conducted at 30°C as follows: strains were thawed on YP + 3% glycerol plates overnight, then patched on YPD plates, and grown overnight. 2 ml YPD liquid cultures were inoculated from patches and grown to saturation by shaking at 30°C, 250 rpm for 21–23 hr ($OD_{600} \approx 7.0$). Cultures were diluted in 50 mL of YP + 1% KOAc to $OD_{600} = 0.25$ and grown overnight by shaking at 30°C, 250 rpm for 15–16 hr. Cells were pelleted and washed once with sterile water, then resuspended in 1% KOAc sporulation media to $OD_{600} \approx 2.5$. Sporulation cultures were shaken at 30°C, 250 rpm and samples were collected every hour for DNA staining and flow cytometry.

## DNA staining for flow cytometry and imaging

Cells were fixed by mixing 0.5 ml of sporulation culture with 1 ml of EtOH. Fixed cells were pelleted and resuspended in water, then sonicated on a Branson Sonifier Model 450 at 10% amplitude for 3 s to break up cell clumps. Cells were pelleted, then resuspended in 100 µl of 40 µg/ml RNase A + 0.05% NP-40 + 50 mM Tris pH 7.4, and incubated at 37°C for 1 hr. Finally, 50 µl of 40 µg/ml proteinase K + 1 µM SYTOX Green + 50 mM Tris pH 7.4 was added to each sample and incubated at 55°C for 1 hr prior to analysis by cytometry.

## Measurement of meiotic divisions and sporulation efficiency

Progression of meiotic divisions was measured by epifluorescence of SYTOX Green-stained cells. 100–200 cells were counted per sample. To measure sporulation efficiency, we counted the proportion of tetrad, dyad, and monad or unsporulated cells from synchronous sporulation cultures after 24–48 hr.

## Protein purification

W303 *S. cerevisiae* strains containing a 2 µm $P_{GAL1}$-kinase-TAP plasmid (pRSAB1234 backbone, originally a gift from Erin O'Shea) were grown overnight to log phase in SC-URA media containing 2% raffinose (Sigma), and then expression of N-terminal kinase domains was induced by addition of 2% galactose (Sigma) for 4 hr at 30°C. Cells were harvested by centrifugation at 8000 rpm, cell pellet washed and resuspended in 1× cell volume of lysis buffer containing 25 mM HEPES pH 8.0, 300 mM NaCl, 0.1% NP-40, 30 mM EGTA, 1 mM EDTA, and a protease/phosphatase inhibitor set was added immediately prior to harvest including 80 mM β-glycerophosphate, 50 mM NaF, 1 mM DTT, 1 mM $Na_3O_4V$, and 1 mM PMSF. The cell slurry was slowly dripped into liquid nitrogen to produce frozen pellets. These pellets were then pulverized in a cryogenic ball mill (Retsch MM301 with 50 ml stainless steel grinding jars) by five rounds of agitation at 15 Hz for 2 min, re-cooling the grinding jars in liquid $N_2$ after each cycle. The grindate was then thawed and cell debris was cleared by centrifugation at 8000 rpm for 30 min followed by sequential filtration through 2.7 and 1.6 µm Whatman GD/X filters (GE). C-terminally TAP-tagged kinases were immobilized on IgG Sepharose 6 Fast Flow beads (GE). These beads (~500 µl slurry per 1 l culture) had been pre-equilibrated in lysis buffer with inhibitors and were then incubated with lysate for 1 hr at 4°C. Bound beads were then loaded into a disposable Bio-Spin column (cat. #732-6008; BioRad) by pipette and washed with 20 ml total wash buffer (lysis buffer + 10% glycerol, 1 mM DTT) at 4°C. The column was then rotated for 20 min at 23°C in 700 µl wash buffer as a final wash to mimic elution conditions. The bound protein was then cleaved from the IgG beads by TEV protease in 600 µl elution buffer (0.21 mg/ml TEV protease

[QB3 MacroLab, UC Berkeley], 25 mM HEPES pH 8.0, 310 mM NaCl, 0.09% NP-40, 26.9 mM EGTA, 0.9 mM EDTA, 1 mM DTT, and 10% glycerol) for 1 hr at 23°C.

For bacterial purification, proteins were expressed with an N-terminal 6× HIS tag in Rosetta2 DE3 pLysS competent cells (QB3 MacroLab, UC Berkeley) by induction with 100 μM IPTG for 18 hr at 16°C. The cell pellet was resuspended in an equal volume of bacterial wash buffer (50 mM Tris-Cl pH 8.0, 300 mM NaCl, 10% glycerol, and 0.1% Triton X-100), freeze-thawed once with liquid nitrogen, then lysed by 3× french press (American Instrument Company) cycles in the presence of 0.05 mg/ml DNAse I, 2 mM MgCl$_2$, 5 mM β-mercaptoethanol, and 1 mM PMSF. Lysate was cleared by centrifugation and filtration as above. NiNTA-Sepharose beads (GE) (~660 μl slurry per 1 l culture) were equilibrated in bacterial wash buffer containing 20 mM imidazole and combined at 1:1 ratio with lysate before adding a further 2.5 mM βME, 1 mM Na$_3$O$_4$V, 80 mM β-glycerophosphate, 50 mM NaF, and 0.5 mM PMSF. This lysate was incubated with beads for batch binding overnight at 4°C. Bound beads were loaded onto a disposable column (as above) and rinsed with the remaining unbound fraction. The column was washed 3 × with 10 ml bacterial wash buffer containing 10 mM imidazole and 2 mM βME, 3 × with 10 ml bacterial wash buffer containing 20 mM imidazole and 2 mM βME, then one final time with 20 ml bacterial wash buffer containing 20 mM imidazole, 2 mM βME, and 2 mM ATP. The column was rotated in previous ATP solution for 15 min at 4°C in an attempt to remove chaperones, before rinsing once more with 5 ml bacterial wash buffer containing 20 mM imidazole and 2 mM βME to remove ATP. The protein was released by incubation with 300 μl of the above buffer containing 250 mM imidazole twice, followed by incubation with 300 μl of the above buffer containing 500 mM imidazole twice. All purification samples were analyzed by SDS-PAGE with Coomassie Brilliant Blue R-250 stain.

## Positional scanning peptide library assays

The library consisted of 182 peptide mixtures having the general sequence Y-A-x-x-x-x-x-S/T-x-x-x-x-A-G-K-K(biotin), where X represents an equimolar mixture of the 17 proteogenic amino acid residues (excluding Ser, Thr, and Cys), S/T indicates an even mixture of Ser and Thr, and biotin is conjugated through an aminohexanoic acid spacer to the C-terminal Lys residue. In each mixture, a single residue at one of the 9 'x' positions was fixed as one of the 20 amino acids. In addition, we included two peptide mixtures in which all 'x' positions were degenerated, but the phosphoacceptor residue was fixed as either Ser or Thr. Peptides were arrayed in 1536 well plates to a final concentration of 50 μM in 2 μl reaction buffer (50 mM HEPES, pH 7.4, 150 mM NaCl, 10 mM MgCl$_2$, 0.1% Tween 20) per well. Reactions were initiated by adding kinase mixed with ATP (final concentration 50 μM with 0.03 μCi/μl [γ-$^{33}$P]ATP). Plates were incubated at 30°C for 2 hr, and then 200 nl aliquots were transferred to streptavidin-coated membrane (Promega), which was quenched by immersion in 0.1% SDS, 10 mM Tris–HCl, pH 7.5, 140 mM NaCl. Membranes were then washed twice with the same solution, twice with 2 M NaCl, and twice with 1% H$_3$PO$_4$, 2 M NaCl. After briefly rinsing with ddH$_2$O, membranes were air-dried and exposed to a phosphor imager screen. Following scanning on a phosphor imager (BioRad), radiolabel incorporation was quantified using QuantityOne software. Data were normalized so that the average signal for a given peptide position was 1. For visualization normalized data from two separate runs were averaged, log transformed, and used to generate heat maps in Microsoft Excel using the color scheme shown in the figures.

## Kinase Peptide assays

Ratiometric specificities were profiled in buffer containing 77.5 mM HEPES pH 7.5, 77.5 mM NaCl, 15.5 mM MgCl$_2$, 250 μM ATP, 0.45 mg/ml BSA, 4.5% glycerol, and 0.2 μCi [γ-$^{32}$P]ATP. Minimal kinase concentrations sufficient for signal were determined empirically and ranged from 5 to 50 nM. Peptides obtained from Tufts University Core Facility (http://www.tucf.com) were added to a final concentration of 45 μM to start the reaction. Comparative peptide assays were always performed in parallel. Reaction assays were aliquoted onto Whatman P81 phosphocellulose (GE) strips, which were then quenched and washed 5 × in 75 mM phosphoric acid to remove free [γ-$^{32}$P] ATP. Samples were dried on a slab gel dryer (Model 1125B; BIORAD) and exposed to a phosphor screen (Molecular Dynamics) to determine the rate of [γ-$^{32}$P] ATP incorporation. Phosphor screens were analyzed with a Typhoon 9400 scanner (Amersham) using ImageQuant software (GE). Final Image quantification was performed using ImageJ (http://

imagej.nih.gov/ij/). Michaelis–Menten curves were generated in a similar manner, except the buffer contained 50 mM HEPES pH 7.5, 50 mM NaCl, 10 mM MgCl$_2$, 500 μM ATP, 83.3 μg/ml BSA, 0.833% glycerol, and 0.2 μCi [γ-$^{32}$P]ATP. In this case, substrate concentration was varied for each kinase peptide combination. Data were fit by nonlinear regression to the Michaelis–Menten model $V_0 = V_{max}*[S]/K_M+[S]$ using Prism (GraphPad software) and Matlab (MathWorks) and this fit was used to determine values for $V_{max}$ and $K_M$.

## Phylogenetic reconstruction of ancestral CMGC kinases

Orthologs of the CMGC gene family were identified by a BLAST search based on the amino acid sequence of *S. cerevisiae* IME2 and *H. sapiens* CDK1, using the NCBI BLAST tool (*Altschul et al., 1990*). To eliminate false positives, hit sequences were reverse BLAST queried, and we eliminated any hits that did not have either IME2 or CDK1 as a result with at least 50% sequence identity. Using the remaining 329 amino acid sequences, a multiple sequence alignment was inferred using MSAProbs with default settings (*Liu et al., 2010*). This alignment was best-fit by the LG model (*Le and Gascuel, 2008*) with a gamma-distributed set of evolutionary rates (*Yang and Kumar, 1996*), according to the Akaike Information Criterion as implemented in PROTTEST (*Abascal et al., 2005*).

Using LG+G, we used a maximum likelihood (ML) algorithm (*Yang et al., 1995*) to infer the ancestral amino sequences with the highest probability of producing all the extant sequence data. Specifically, we used RAxML version 7.2.8 to infer the ML topology, branch lengths, and evolutionary rates (*Stamatakis, 2006*). We exported this ML phylogeny to another software package, PhyML (*Guindon et al., 2010*), in order to calculate statistical support for branches as approximate likelihood ratios. We next reconstructed ML ancestral states at each site for all ancestral nodes using the software package Lazarus (*Hanson-Smith et al., 2010*). We used sequences from the CK family as the outgroup to root the tree. We placed ancestral insertion/deletion characters according to Fitch's parsimony (*Fitch, 1971*), in which each indel character was treated independently.

We extracted the ancestral sequences from the phylogenetic nodes corresponding to AncCMGI, AncLF4, AncNgru, AncICK, AncCneo, AncNcra, and AncYlip. We characterized the support for these ancestors by binning their posterior probabilities of states into 5%-sized bins and counting the proportion of ancestral sites within each bin (*Figure 2—figure supplement 1*). We also generated alternate versions of these ancestral sequences by randomly sampling from their posterior distributions to generate between 2 and 3 alternate ancestors for every node, as described in *Williams et al. (2006)*.

## Robustness to uncertainties in sequence reconstruction

Ancestral sequence reconstruction is a probabilistic method and involves uncertainties in the amino acid identities. Even for relatively well-conserved protein families like kinases, these uncertainties become more pronounced when attempting to reconstruct deep ancestors. A summary of the statistical supports for each of the resurrected kinases is presented in *Figure 2—figure supplement 1*. For each of the seven resurrected kinases between 60 and 90 alternative amino acids were sampled by a Bayesian method to address the impact of uncertainties on our results. In all cases, both the overall primary specificities and the +1 specificities as determined by individual peptides were robust to uncertainties in ancestral sequence reconstruction. That is, kinases that we resurrected with amino acid substitutions that explored different possible amino acids had very similar activity levels and specificities as the maximum likelihood ancestors presented in the main figures. The degree of preference for proline varies slightly in the deepest alternative reconstruction, but the general trend of evolution from +1P preference, to expanded specificity, and finally to +1R preference, is maintained. The data from alternative reconstructions is presented in *Figure 2—figure supplement 2* and *Figure 3—figure supplement 2*. Together, these results give us confidence in the validity of our approach.

## Acknowledgements

The authors thank Elçin Ünal for yeast strains and advice on meiosis protocols; Lillian Fritz-Laylin, Louise Glass, Alexander Bullock, Zheng Fu and Tom Sturgill for DNA; Rune Linding and Pau Creixell

for sharing results prior to publication; Doug Koshland, members of the Koshland lab for discussion of the project; and Woj Wojtowicz, Carlos Pantoja, Vincent Guacci, William Ludington, and David Morgan for critical reading of the manuscript. Work in the Johnson laboratory was supported by NIH grant GM037049. Work in the Turk laboratory was supported by NIH grants R01 GM105947 and R01 GM104047, and CJM was supported by training grant T32 GM007324. Work in the Holt laboratory was supported by the William Bowes Research Fellowship.

## Additional information

### Author contributions

CJH, VH-S, KJK, BET, LJH, Conception and design, Acquisition of data, Analysis and interpretation of data, Drafting or revising the article; CJM, Acquisition of data, Analysis and interpretation of data, Drafting or revising the article; HJL, Acquisition of data, Analysis and interpretation of data; ADJ, Analysis and interpretation of data, Drafting or revising the article

### Funding

| Funder | Grant reference number | Author |
|---|---|---|
| University of California Berkeley | Bowes Research Fellows Program | Conor J Howard<br>Kristopher J Kennedy<br>Liam J Holt |
| National Institute of General Medical Sciences | R01 GM105947 | Chad J Miller<br>Hua Jane Lou<br>Benjamin E Turk |
| National Institute of General Medical Sciences | R01 GM104047 | Chad J Miller<br>Hua Jane Lou<br>Benjamin E Turk |
| National Institute of General Medical Sciences | GM037049 | Victor Hanson-Smith<br>Alexander D Johnson |

The funders had no role in study design, data collection and interpretation, or the decision to submit the work for publication.

### Author ORCIDs

Liam J Holt, http://orcid.org/0000-0002-4002-0861

## Additional files

### Supplementary files

• Supplementary file 1. List of plasmids generated in this study.

• Supplementary file 2. List of yeast strains generated in this study.

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
