## [Decision Letter]

Thank you for sending your work entitled “Ancestral resurrection reveals evolutionary mechanisms of kinase plasticity” for consideration at eLife. Your article has now been evaluated by Michael Marletta (Senior editor), a Reviewing editor, and 2 reviewers, one of whom is also a member of the Board of Reviewing Editors. As you will see (below) the reviewers thought the work was original and interesting. It is really impressive that one can generate functional protein kinases from maximum-likelihood reconstructions of ancestral sequences, and we suspect that this approach will continue to yield interesting insights into protein evolution. However, they also had one major, substantive issue with the paper's interpretations, which would need to be accommodated in a revised version.

The Reviewing editors and the other reviewer discussed their comments before we reached this decision, and the Reviewing editor has assembled the following comments to help you prepare a revised submission.

1) The main problem is the assertion that the ancestral GMGI kinase (AncCMGI) kinase had a strong +1 proline preference, and then this specificity broadened to Pro or Arg, and then finally to Arg, as the fungal IME2s evolved. The Pro preference for AncCMGI is asserted rather unequivocally (“The ancestor of all CMGC kinases had strong specificity for proline at its +1 site”).

However, data in Figure 3 actually show pretty modest preference for serine peptides and a bit better for threonine peptides, but neither is as strong a preference as extant proline-directed kinases are for proline. The very modest preference for Pro in the Ser-peptides is particularly problematic given that most phosphorylations occur at Ser residues. Moreover, one of the “alternative” versions of AncCMGI shown in Figure 3—figure supplement 2 had even more modest preference for a +1 proline than the version characterized in the main figures. Thus, by your own data, the ancestral kinase was rather “purple” compared to the “blue” S. cerevisiae IME2, the “red” ERKs, JNKs, SAPKs, and CDKs.

This problem is compounded by the fact that assignments of nodes in the tree are less certain near the base of the tree than near the leaves. For example, it is possible that your (purple) AncNgru protein is really closer to the root of the tree than the (reddish-purple) AncCMGI protein is.

This is not a problem that needs to be addressed with further experiments or phylogenetic reconstructions, but rather with substantial re-writing of all parts of the paper, acknowledging that your findings are consistent with two plausible evolutionary schemes: one where the ancestral kinase is relatively broad in its +1 specificity, and one where it first becomes broader and then becomes narrower. So in your Figure 6, either the left-hand species, as you suggest, or the middle species could be the ancestral state.

---

## [Author Response]

1) The main problem is the assertion that the ancestral GMGI kinase (AncCMGI) kinase had a strong +1 proline preference, and then this specificity broadened to Pro or Arg, and then finally to Arg, as the fungal IME2s evolved. The Pro preference for AncCMGI is asserted rather unequivocally (“The ancestor of all CMGC kinases had strong specificity for proline at its +1 site”).

However, data in Figure 3 actually show pretty modest preference for serine peptides and a bit better for threonine peptides, but neither is as strong a preference as extant proline-directed kinases are for proline. The very modest preference for Pro in the Ser-peptides is particularly problematic given that most phosphorylations occur at Ser residues. Moreover, one of the “alternative” versions of AncCMGI shown in Figure 3—figure supplement 2 had even more modest preference for a +1 proline than the version characterized in the main figures. Thus, by your own data, the ancestral kinase was rather “purple” compared to the “blue” S. cerevisiae IME2, the “red” ERKs, JNKs, SAPKs, and CDKs.

This is true. We have substantially revised the tone of the article in response to this comment and acknowledged the two alternative possibilities in the manuscript. Our original contention that AncCMGI was specific for proline came from the PSPL screens for the ML reconstruction, which revealed a strong +1 proline signal (Figure 2). However, unlike true “proline-directed” kinases (i.e. cyclin-dependent kinases or MAP kinases), the PSPL data do show a clear arginine signal at the +1 position, indicating that the AncCMGI did have the capacity to phosphorylate +1 Arg substrates. Furthermore, as pointed out by the reviewers, assays with consensus peptide substrates (Figure 3) show an even more modest preference for proline. We would point out that the consensus peptide assays could underestimate the influence of the +1 position due to the sequence context, perhaps by fixing the most preferred residues at the -3 and -2 positions. While it is also correct that the different reconstructions of AncCMGI vary in their proline preference, the +1P/R ratio for all of the AncCMGI reconstructions is greater than the +1P/R ratio for all of the AncNgru reconstructions. These data suggest strongly that +1 specificity did indeed broaden with evolution along the CMGI lineage.

We have addressed this major concern through revising the manuscript in several ways to reduce the emphasis on +1 proline preference in our ancestor. Specifically, in the Results narrative our description of AncCMGI is more nuanced, indicating quantitatively the proline preference from PSPL data (5.6-fold), and emphasizing that this preference is more modest than for extant CDKs and MAPKs. We also refer to differences between the three AncCMGI reconstructions in proline selectivity, and include a new panel in Figure 3—figure supplement 2 that shows +1 position PSPL data for multiple reconstructions of each ancestor.

In addition, we have updated our colors in Figure 3 to reflect the modest proline specificity of AncCMGI.

This problem is compounded by the fact that assignments of nodes in the tree are less certain near the base of the tree than near the leaves. For example, it is possible that your (purple) AncNgru protein is really closer to the root of the tree than the (reddish-purple) AncCMGI protein is.

This is extremely unlikely. When using the best-fitting maximum likelihood phylogenetic model (PROTCATLG), the placement of AncNgru as a descendant of AncCMGI is 6.41x10^5 times more likely than the alternative hypothesis in which the Ngru clade (containing LF4 and MOK sequences) is basal to the CMGI clade. Although the true evolutionary history of these proteins cannot be directly observed, the sequence evidence in this case provides overwhelming support for a model in which AncNgru is a descendant of AncCMGI. To put this more intuitively: AncNgru gives rise to only the LF4/RCK/IME2 kinase family while AncCMGI gives rise to the LF4/RCK/IME2, CDK, MAPK, CDKL, CKL, and GSK families. From this standpoint, it is most appropriate to place AncCMGI basal to AncNgru.

This is not a problem that needs to be addressed with further experiments or phylogenetic reconstructions, but rather with substantial re-writing of all parts of the paper, acknowledging that your findings are consistent with two plausible evolutionary schemes: one where the ancestral kinase is relatively broad in its +1 specificity, and one where it first becomes broader and then becomes narrower. So in your Figure 6, either the left-hand species, as you suggest, or the middle species could be the ancestral state.

We agree, and have made this point throughout the text and the Discussion. As described above, our data are not unequivocal regarding the specificity of the deep ancestor, which could phosphorylate +1 arginine to some extent. For example, the Discussion now reads:

“Taken together with the measured specificities of reconstructed CMGC kinases, these results suggest that the evolutionary trajectory from +1P to +1R passed through a historic ancestor with hybrid +1 PR specificity. However, due to the modest degree of +1 specificity in AncCMGI (5.6-fold by PSPL and 2-fold in the context of an otherwise perfect consensus peptide) together with slight fluctuations in degree of specificity in alternate reconstructions, an alternative possibility is that AncCMGI was a broad specificity kinase that subfunctionalized to modern proline or arginine specific kinases.”

With respect to Figure 6, we have updated the Figure 6 title to “Simplified schematic of one possible path of kinase network evolution” to make it clear that we are simply exemplifying our discussion with a visual example that helps the reader think about the experiment in Figure 5 where we know for certain that we have expanded the specificity of the extant Ime2 kinase.